# Faster and Cheaper Energy Demand Forecasting at Scale

## Abstract

Energy demand forecasting is one of the most challenging tasks for grids operators. Many approaches have been suggested over the years to tackle it. Yet, those still remain too expensive to train in terms of both time and computational resources, hindering their adoption as customers behaviors are continuously evolving. We introduce Transplit, a new lightweight transformer-based model, which significantly decreases this cost by exploiting the seasonality property and learning typical days of power demand. We show that Transplit can be run efficiently on CPU and is several hundred times faster than state-of-the-art predictive models, while performing as well.

## 1 Introduction

Energy demand forecasting is of utmost importance for grid operators and energy producers to ensure the stability of their service. Over the past decades, many approaches to provide the most accurate forecasting have been suggested [10]. In the recent years, approaches based on transformers models have started to be explored, but the cost to train them at scale, in a context where frequent retraining is a norm rather than an exception, hinders their adoption [4, 3]. Indeed recent events (e.g. climate change, remote working, energy prices) have highlighted that the way we consume electricity is in constant evolution and models need to be regularly retrained to adapt to new contexts. This makes previously suggested methods hardly applicable and therefore calls for cheaper and faster models.

Hence, we introduce Transplit, a transformer-based approach trained to recognize typical days of consumption from various profiles, that requires less computation time and resources, while conserving state-of-the-art performances. To validate our approach, we compare it against 4 related state-of-the-art transformers based approaches on two different datasets of electricity consumption: a publicly available one and another from our industrial partner (a national grid operator), that cannot be shared for privacy reasons. Overall, we observe that our approach performances are in line with the best approaches, while being **380** to **940** times faster to train on a single CPU, without relying on dedicated resources such as GPU, allowing it to run on significantly cheaper platforms.

## 2 Forecasting Energy Demand

Load forecasting is essential for electricity grid management and security, with applications ranging from peak prediction, incident prevention, to maintenance planning. It usually involves to forecast the energy demand of a household or a neighborhood in kWh over a given period of time (from minutes to months) based on historical data. Early on, Machine Learning approaches have been suggested [2], but their application remained limited by the amount of available data and computing power, which is not an issue anymore with the development of smartgrid and the recent increase in computing capability. Among ML-based approaches, we distinguish two categories.

Submitted to 36th Conference on Neural Information Processing Systems (NeurIPS 2022). Do not distribute.

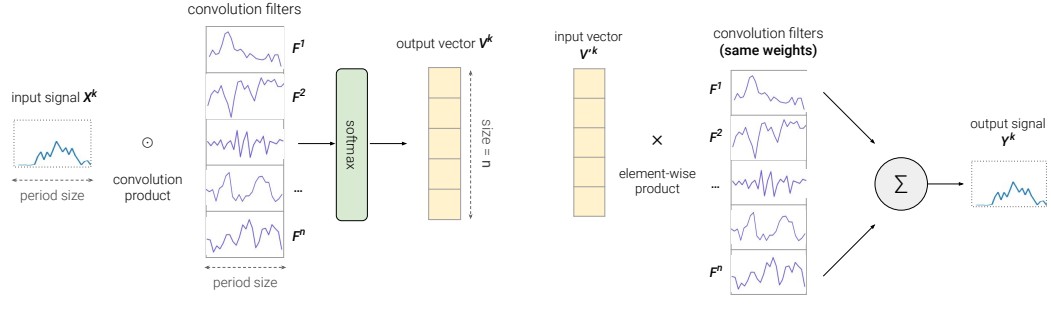

(a) The Slice2Vec process.       (b) The Vec2Slice process.

Figure 1: The SVS module is composed of two layers which share the same weights.

Approaches falling in the first one consist in training one model per household. This was usually achieved by models such as random forests, SVMs, or linear regressions [11][3], but more recently neural networks based approaches started to be suggested. Neural Prophet (2021) [15] for example has been applied to electric load forecasting [14] with interesting results, as well as Deep Probabilistic Koopman models [9]. Although these approaches perform well, the number of models to train for a grid rapidly becomes too expensive to deploy.

On the opposite, approaches from the second category rely on a single model which generalizes different behaviors of the demand time series, discerns consumers' profiles and prolong their demand curve according to their shape and other factors. Those approaches are usually less costly, but also less precise. LSTMs [5] have been widely used in this context [12][18] and are still the subject of recent load forecasting research [7][18].

Yet, with the development of transformers [16], approaches such as LogTrans [8], Reformer [6], Informer [19] and lastly Autoformer [17] started to appear with better accuracy and efficiency. These approaches all derived from the original work differ in their inner attention mechanism that is modify to lower the computation complexity. Still, these models are not entirely applicable to energy demand forecasting as they require high computation power and considerable time to train, which is problematic when models need to be retrained often. There is therefore a need for a much lighter model, requiring less computational power and time, with at least similar performances.

## 3 Proposed approach

In this regards, we introduce Transplit, a lightweight load forecasting model. To design it, we started from an observation on a fundamental difference between most of language models and time series models, which also impacts their efficiency.

### 3.1 Chunking time series

Whereas language models perform well by tokenizing input text [13] instead of splitting it character by character, current time series models process every single input as a vector. As electrical consumption data is seasonal, we propose to embed each season – in our case study, the time series is split into days. The encoding and decoding parts are achieved by learning a vocabulary of daily consumptions. Our solution then consists in reasoning in blocks of consumption days rather than single values.

#### 3.1.1 Slice2Vec

As described in Figure 1a, Slice2Vec takes a sequence $X \in \mathbb{R}^L$, where $L = m \times T$ is the input sequence length which is a multiple of a period $T$. $X$ is split into $m$ periods of size $T$, that we denote $(X^k)_{k \in \{1,...,m\}}$[1].

For each $X^k$, we define a vector $V^k$ with the following convolution product:

$$\forall i \in \{1, \ldots, n\}, V_i^k = X^k \odot F^i \tag{1}$$

---

[1]Concretely, if the period $T$ is one day, then $X^k$ is the $k^{th}$ day of the input.

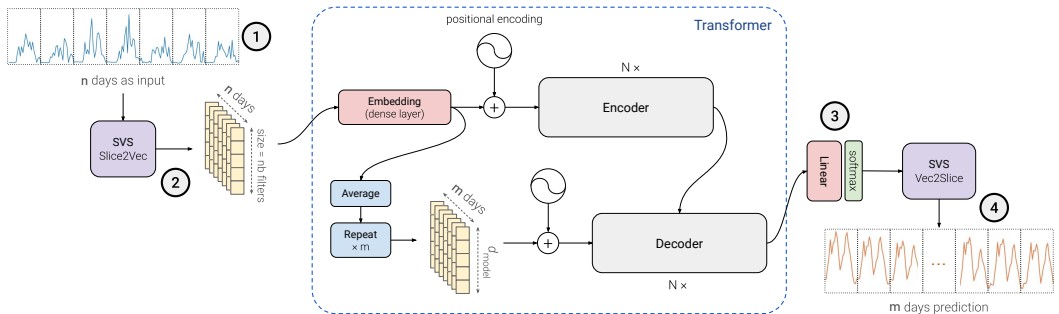

Figure 2: Model architecture of Transplit: (1) the input consumption is split into days; (2) Slice2Vec converts each day into a vector and passes it to the transformer (3) that forecasts vectors (4) which are converted to daily consumptions with Vec2Slice.

where $n$ is the chosen size for the vectors, $V_i^k$ is the $i^{th}$ coefficient of $V^k$, and $F^i$ is a convolution filter of size $T$. Since $X^k$ and $F^i$ have the same size, the convolution product is also equivalent to the sum of their term-wise product.

### 3.1.2 Vec2Slice

In order to transform a vector $V'^k$ back in a $T$-sized slice of time series called $Y^k$, the filters $F^i$ are weighted according to the coefficients of $V'^k$, then summed:

$$Y^k = \sum_i V_i'^k F^i \tag{2}$$

as illustrated in Figure 1b. The slices $Y^k$ are concatenated to form the output $Y$. It is important to underline that Vec2Slice *is not* the inverse function of Slice2Vec: $Vec2Slice(Slice2Vec(X)) \neq X$.

Vec2Slice shares the filters weights with Slice2Vec. The filters are therefore used twice: to recognize the input *and* to match the shape of the expected output.

## 3.2 Transplit

From these two components, we propose *Transplit*, a new load forecasting model based on the original transformer [16] with simple full attention mechanisms. The architecture of the model is presented in figure 2.

# 4 Evaluation

## 4.1 Criteria

Transplit is designed to (**req. 1**) maintain state-of-the-art performances for load forecasting, while (**req. 2**) being lighter and faster, in terms of number of parameters and training time.

To validate these two requirements, we compare Transplit with 4 state-of-the-art approaches: Autoformer (2021) [17], Informer (2021) [19], Reformer (2020) [6] and a vanilla Transformer architecture (2017) [16]. Classical models such as LSTMs are disregarded as it has been shown that transformers perform better for load forecasting [17, 19, 6]. The assessment is done on two forecasting ranges: 7 days and 30 days. We use two datasets and two performance metrics, and also measure training time taken by using a GPU and only using a CPU (without multiprocessing).

## 4.2 Experimental setting

We evaluate those approaches on 2 datasets of household's hourly consumption expressed in kWh. The first one, denoted **IND**, is provided by our industrial partner (a national grid operator) and encompasses the consumption of 6010 households over 2 years (2020-2021). The second one **ECL**[1] is an open dataset containing the consumption of 321 households over 3 years (2012-2014).

|  | ECL | | | | IND | | | |
| --- | --- | --- | --- | --- | --- | --- | --- | --- |
|  | 7 days | | 30 days | | 7 days | | 30 days | |
|  | MSE | MAE | MSE | MAE | MSE | MAE | MSE | MAE |
| **Transplit** | 0.584 | 0.178 | **0.688** | 0.189 | 0.177 | 0.189 | **0.202** | **0.205** |
| **Autoformer** | **0.577** | **0.153** | 0.736 | **0.174** | 0.185 | 0.210 | 0.204 | 0.217 |
| **Informer** | 2.68 | 0.376 | 2.71 | 0.387 | 0.201 | 0.213 | 0.226 | 0.231 |
| **Reformer** | 1.28 | 0.262 | 1.79 | 0.313 | **0.172** | **0.185** | 0.205 | 0.207 |
| **Transformer** | 0.781 | 0.197 | 0.918 | 0.213 | 0.177 | 0.191 | 0.203 | 0.207 |

Table 1: Error metrics for the ECL and IND datasets

|  | **Transplit** | **Autoformer** | **Informer** | **Reformer** | **Transformer** |
| --- | --- | --- | --- | --- | --- |
| **# of parameters** | **183,616** | 10,505,217 | 11,306,497 | 5,782,529 | 10,518,529 |
| **ECL**: train. with **GPU** | **3m02s** | 4h16m | 1h53m | 4h13m | 2h51m |
| **IND**: train. with **GPU** | **21m13s** | 1d 12h15m | 19h37m | 20h56m | 1d 9h00m |
| **ECL**: train. with **CPU** | **2m47s** | 1d 18h01m | 17h11m | 1d 13h49m | 1d 3h01m |
| **IND**: train. with **CPU** | **32m05s** | 20d 21h59m | 8d 13h21m | 18d 19h47m | 13d 10h42m |

Table 2: Number of trainable parameters and training time for each model, dataset and infrastructure used, for a $720 \rightarrow 720$ values forecast.

We set ratios of training / validation / testing to respectively 70 / 10 / 20%, based on the timeline. We feed the models with inputs of 720 values (30 days).

We configure all baseline approaches according to the recommendation from their authors for the ECL dataset. Regarding Transplit, we set $T = 24$ hours and 512 filters for the SVS module and use 1 encoder and 1 decoder, with a vector size ($d_{model}$) of 64.

While the training data is standardized, we reverse it at the output in the testing phase to reflect the loss with the original scale in kWh. Further details are available on our repository[2].

## 4.3 Results

Results for both datasets are presented in Table 1. Although there is no significant improvement, Transplit succeeds in keeping the performance of state-of-the art time series transformer models.

The key element of Transplit is its lightness: what is important to distinguish is the time taken to train these models, shown in Table 2. We observe that Transplit took 21 minutes to learn the 6010 consumers profiles of the IND dataset using a GPU, and is then 102 times faster than the Autoformer.

Another big advantage for Transplit is that using a CPU instead of a GPU doesn't significantly slow down the model's computation time, as processing smaller sequences implies smaller operations, and data still has to be cached in the GPU, which can take a lot of time in total. The non-necessity of GPU to efficiently run the model allows it to be used on less expensive infrastructures, while being faster and conserving good forecasting performances.

## 5  Conclusion

We have seen that Transplit remains in line with state-of-the-art time series deep learning models that are able to generalize consumption profiles and predict a detailed state of the grid in terms of electric load, while being several hundreds of times faster and lighter.

It opens up the possibility to run a deep-learning predictive model globally at a lower cost, as only one CPU is enough to keep Transplit efficient, making it possible to run it locally. This finally offers the possibility to update the model regularly with new data. Yet, Transplit might be ineffective to predict non-seasonal time series, but could be extended to seasonal signals in general.

---

[2]https://anonymous.4open.science/r/Transplit-BC41

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
