# OpenReview forum: "Faster and Cheaper Energy Demand Forecasting at Scale"
_NeurIPS.cc/2022/Workshop/HITY — HITY Workshop NeurIPS 2022_

### Official Review · Reviewer_uaNt · 2022-10-06
**Light-weight model for energy demand forecasting**

**Rating:** 1
**Confidence:** 3

**Review:**

The paper proposes a transformer-based model, which is light-weight and much faster to train. The results look promising and the text is reasonably clear.

---

### Official Review · Reviewer_W6aq · 2022-10-16
**Accept: The proposed method seems to improve the feasibility of energy demand forecasting using neural networks**

**Rating:** 1
**Confidence:** 3

**Review:**

The paper proposes a simple transformer architecture, called Transplit, for efficient energy demand forecasting. It is based on chunking the input sequence before passing it into an encoder-decoder transformer model. By using much less parameters than alternative methods, it is signigicantly faster and cheaper to train, while achieving comparable performance to the relevant baselines.

Since the proposed method seems to improve the feasibility of using neural network based models for energy demand forecasting, I recommend to accept the paper.

---

### Decision · Program_Chairs · 2022-10-20

Accept